# Juvenile cleaner fish can socially learn the consequences of cheating

Noa Truskanov [1✉], Yasmin Emery [1] & Redouan Bshary [1]

Social learning is often proposed as an important driver of the evolution of human cooperation. In this view, cooperation in other species might be restricted because it mostly relies on individually learned or innate behaviours. Here, we show that juvenile cleaner fish (*Labroides dimidiatus*) can learn socially about cheating consequences in an experimental paradigm that mimics cleaners' cooperative interactions with client fish. Juvenile cleaners that had observed adults interacting with model clients learned to (1) behave more cooperatively after observing clients fleeing in response to cheating; (2) prefer clients that were tolerant to cheating; but (3) did not copy adults' arbitrary feeding preferences. These results confirm that social learning can play an active role in the development of cooperative strategies in a non-human animal. They further show that negative responses to cheating can potentially shape the reputation of cheated individuals, influencing cooperation dynamics in interaction networks.

¹ Institute of Biology, University of Neuchâtel, Rue Emile-Argand 11, 2000 Neuchâtel, Switzerland. ✉email: noatrs@gmail.com

Social interactions often involve conflicts of interest between participants[1–3], providing opportunities for both cooperation and defection. These interactions frequently occur in the presence of observing individuals, who can gain valuable information about the costs and benefits associated with different strategies, and the nature of potential interaction partners[4–6]. While social learning is often suggested to be vital to the evolution of large-scale cooperation in human societies[7–10], its role in shaping cooperation dynamics in other species is currently unclear[5,11]. Although cooperation and social learning are widespread in nature[12–15], evidence for the use of social learning by non-human animals in cooperative contexts is extremely limited[6,16,17]. This, in turn, often leads to the assumption that social learning about cooperative behaviours is a uniquely human feature[11,18].

Studies aiming to test the potential influence of social learning on cooperation in other animals, must bear in mind that even in humans, the links between the two are ambiguous and highly contentious[5,7–9,19–23]. Observation of the social behaviour of others may provide individuals with information on different aspects of cooperative interactions, including the behavioural strategies being employed, their prevalence, associated payoffs and consequences. Social learning can thus promote either more helping, or exploitation, depending on the information being used and the biases that underlie its acquisition (also termed social learning strategies, i.e., what and from whom to learn, and under which circumstances[24,25]). Identifying the pathways of information transmission is crucial for determining the impact of social learning on cooperative interactions and their evolutionary stability.

Learning socially to adjust cooperative behaviours can be especially beneficial to animals living in complex social environments, involving multiple interaction partners, or different potential behavioural strategies. In such cases, individuals are faced with the challenge of tracking the characteristics of partners and identifying the behavioural strategies that are expected to be beneficial in interactions with them. While individual learning provides a more direct way to assess the consequences of different social strategies, it can be costly, time-consuming and lead to errors in cases in which personal experience is limited[24–26]. Taking into account social information, gained by observing the interactions of others, can help individuals bypass these problems, and increase the probability that they will discover the optimal social strategies in different interaction contexts.

A model system in which such learning might be particularly beneficial is the mutualistic interactions of the bluestreak cleaner wrasse (Labroides dimidiatus) with a great diversity of so-called 'client' reef fish. Cleaners offer clients cleaning services in which they remove the clients' ectoparasites[27]. However, a conflict of interests arises, as cleaners prefer to feed on clients' protective mucus, which is costly to the clients and constitutes cheating[28]. Clients use various control mechanisms (responses that reduce cheaters' payoffs[29]) in response to cleaners' cheating: they can either leave the cleaning station, or terminate the interaction by chasing the cleaners, which functions as punishment[30]. These responses vary between different client species, and are important for enforcing cooperation, causing the cleaners to eat against their preference[31]. Given that cleaners interact with a range of clients that differ in abundance, parasite load, mucus quality and responsiveness to cheating[32–34], observing conspecifics interact with clients may provide cleaners with useful information about when and with whom to cooperate. Such information may be especially relevant to young cleaners, who tend to spatially overlap with adults, but may have had less opportunities to gain relevant personal experience as their client assembly shifts along their ontogeny[35,36].

Here, we experimentally test the ability of juvenile cleaners to learn socially from observing the interactions of adults with unfamiliar model clients. To simulate the key features of natural cleaner–client interactions, we use a well-established experimental paradigm that has already contributed substantially to the study of cooperation (e.g., [37,38]). We present cleaners with plexiglass plates (model clients) containing both prawn and fish flake items on their surface. As cleaners generally prefer prawn to fish flakes, eating flake items requires them to eat against this preference, and corresponds to behaving cooperatively by eating ectoparasites in the wild. Eating a preferred prawn item, on the other hand, represents eating client mucus, which constitutes cheating in natural interactions (also see[31,37,38]). The juveniles can observe interactions in which adult cleaners are presented with model clients and are free to choose which items to consume, but their choices would elicit pre-defined responses (the plates are attached to levers, enabling the experimenter to control their movement). Behaving 'cooperatively' by eating against preference causes the plate to remain available for longer, whereas 'cheating' by consuming a preferred prawn item elicits an evasive response by the plate, equivalent to clients' reactions to cheating in the wild and prohibiting further food consumption.

Using this setup, we conduct three experiments in which we test whether young cleaners can learn socially about different aspects of cooperative interactions with clients. We test whether the cleaners: (1) behave more cooperatively by eating more against their preference, following observation of adults interacting with model clients that flee in response to cheating (the consumption of preferred food); (2) prefer to service model clients that were observed responding more favourably to conspecifics' cheating and (3) copy adults' arbitrary, non-meaningful, model client preferences. We find that juvenile cleaners can learn socially to adjust their cooperative behaviour and client choice to model clients' responses, but do not copy adults' arbitrary preferences. Our results thus show that social learning about the consequences of cheating can shape strategic behaviour in young cleaner fish, and suggest that social learning may play an active role in the maintenance of cooperative interactions in natural model systems.

## Results

**Socially learning to feed against preference.** In the first experiment, we tested whether naive juveniles would socially learn to eat against preference after observing adults interacting with plates that leave in response to the consumption of preferred food. In the observer treatment, juveniles could observe the adult interacting with the plate, and the plates' subsequent responses (rapid 'fleeing' when prawn is consumed). In the control treatment, no interaction between the adult and the plate could occur, but the plates were rapidly removed after being presented for a similar amount of time (Fig. 1). For all tested individuals, we first measured their baseline preference for prawn, and then tested their feeding choices following observation in a set of 15 consecutive tests, in which plate responses were similar to those of the observation phase. The design thus allowed subjects to combine the social information with personal experience obtained during the tests, a situation that is likely to resemble juvenile cleaners' opportunities for social learning under natural conditions.

Our results reveal clear evidence for social learning: juvenile cleaners that could observe adults interacting with simulated clients, ate more against their preference than individuals in the control treatment who did not observe the social interaction itself (LM: social observation: estimate $\pm$ SE = 0.339 $\pm$ 0.114, $N = 20$, $F = 8.796$, d$f = 1$, $P = 0.009$, effect size (Cohen's $d$) = 1.322,

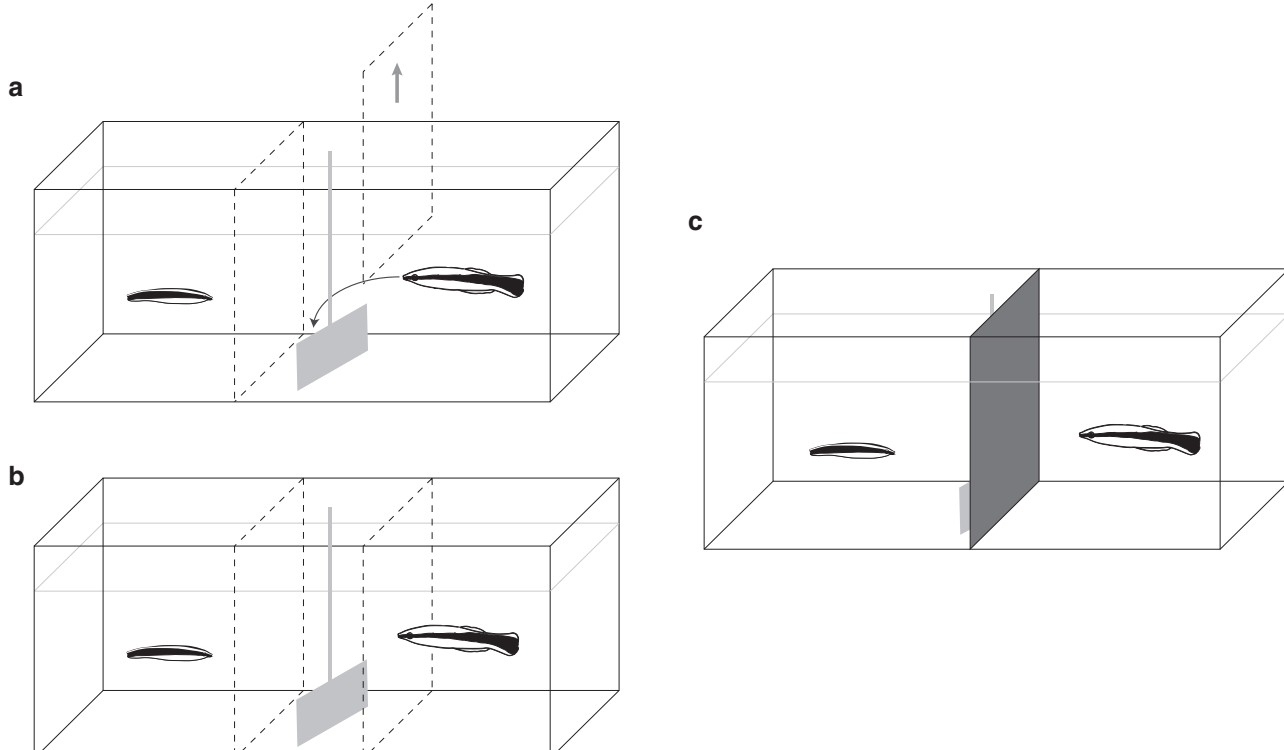

**Fig. 1 Experimental setup of experiments 1 and 2.** During the demonstration phase (**a**, **b**), the cleaners were separated via a clear partition (juvenile depicted on the left side, and as smaller compared to the adult). In each trial, a plate with food facing the juvenile was introduced behind a second clear partition that made it inaccessible for the adult. **a** In the observer treatments, the clear partition between the demonstrator and the plate was removed, and the demonstrator was allowed to interact with the plate. **b** In the control treatments, the plate remained confined and no interaction could occur. **c** During individual testing, an opaque barrier separated the young and adult cleaners. The juveniles were initially confined and could then approach the plates presented. The plates would follow the same response rules they exhibited during training.

Fig. 2). Thus, the potential for individual experience in the tests was not sufficient to override the advantage of being initially exposed to social information. Flake palatability (i.e., the experiment was conducted in two cohorts differing in the palatability of flake items) did not seem to affect the cleaners' feeding choices (LM: flake palatability: estimate ± SE = 0.12 ± 0.114, $N = 20$, $F = 1.097$, d$f = 1$, $P = 0.309$). Importantly, the results of this experiment indicate that social observation can affect cleaners' feeding in ways that correspond to behaving more cooperatively under natural conditions.

**Socially learning about partner responsiveness.** In the second experiment, we tested whether juvenile cleaners can use social learning to assess client responsiveness and prioritise plates that are tolerant to cleaners' consumption of preferred food over plates that negatively respond to it (after verifying that the subjects could also learn this through individual learning, for further details see Supplementary Note 1 and Supplementary Fig. 2). Each juvenile cleaner experienced four different treatments (treatment order counterbalanced): in the two 'observer' treatments, juveniles could observe an adult interacting separately with a tolerant and a responsive plate, differing in colours and patterns. In these interactions, the responsive plate would respond to cleaners' 'cheating' by either leaving swiftly (fleeing) or chasing the cleaner for ~3 s (punishing). In the two control treatments, the adults were blocked from interacting with the plates by a transparent barrier but both observation time and plate movement (gentle leaving/fast fleeing/simulated punishing) were matched to the observer treatments (Fig. 1). At the end of each treatment, the juveniles' preferences were tested in a set of ten choice tests in which the chosen plate's responses to the

consumption of prawn would match the ones used during the demonstration phase. Thus, like in experiment 1, the juveniles could potentially combine the social information with personal experience gained during this test phase.

We found that social observation influenced juveniles' model client choice (GLMM: social observation: estimate ± SE = 0.345 ± 0.208, $N = 19$, $\chi^2 = 15.665$, d$f = 1$, $P < 0.0001$, Fig. 3) regardless of plate reaction type (GLMM: effect of plate response type: estimate ± SE = −0.256 ± 0.207, $N = 19$, $\chi^2 = 0.011$, d$f = 1$, $P = 0.915$; plate response type × social observation: estimate ± SE = 0.497 ± 0.297, $N = 19$, $\chi^2 = 2.789$, d$f = 1$, $P = 0.095$; $R^2 = 0.04$). Observing adult cleaners interacting with the plates, diverted juveniles' preferences towards tolerant model clients: only following trials of the 'observer' treatments the cleaners significantly preferred the tolerant plates, whereas after experiencing control conditions, their preferences remained at chance level (GLHT comparing the juveniles' preferences to a no preference null hypothesis: observation of fleeing, $z = 2.472$, $P = 0.027$; observation of punishing, $z = 3.889$, $P = 0.0002$; control fleeing, $z = 0.275$, $P = 0.783$; control punishing, $z = −1.378$, $P = 0.168$, Fig. 3). As in experiment 1, the effect of social learning in this experiment remained pronounced despite the potential for individual learning during the tests. Taken together, our results confirm that observer juvenile cleaners can extract information about plates' reactions that correspond to clients' partner control mechanisms in nature and use this information to guide their choices.

**Observation of adults exhibiting arbitrary preferences.** In the third experiment, we tested whether juvenile cleaners would copy whichever behaviours they happen to observe, by exposing them to social information regarding adult cleaners' arbitrary feeding

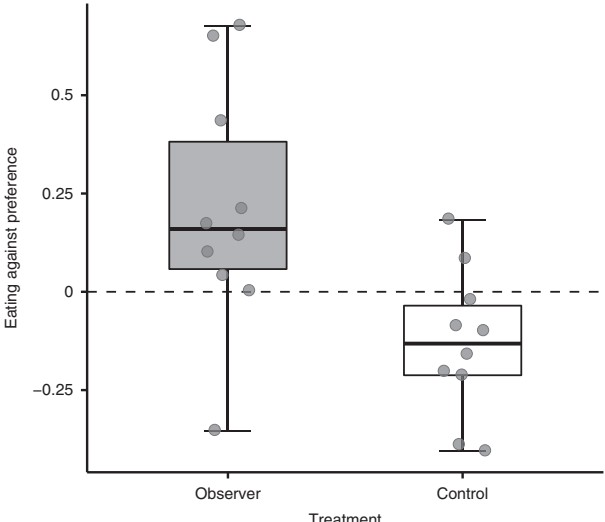

**Fig. 2 Socially learned adjustment of a cooperative foraging decision.** The effects of social observation on the extent to which juvenile cleaners ate against preference in the test phase of experiment 1. Dashed line marks the juveniles' predicted score based on their initial feeding preferences. Values above and below 0, indicate that the fish ate more and less flake items, respectively, than predicted according to their baseline food preferences. Observer treatment, in which the juveniles had witnessed the plates fleeing in response to adults' consumption of preferred prawn, is marked in grey ($N = 10$). Control treatment, in which the adults could not interact with the plates, is marked in white ($N = 10$). Boxplots show the median and interquartile range, whiskers denote 1.5 × interquartile range and dots mark individual data points. Source data are provided as a Source Data file.

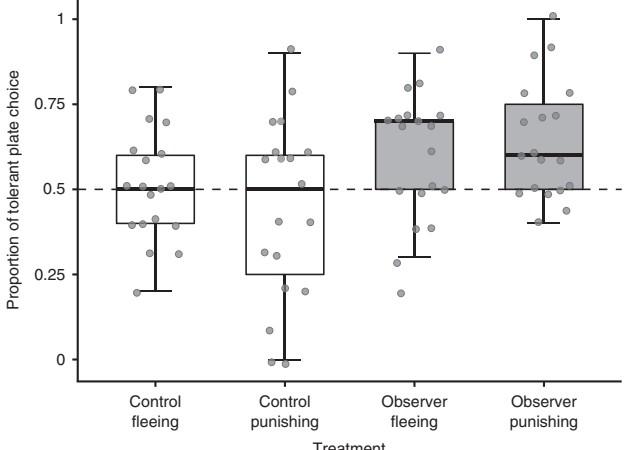

**Fig. 3 Socially learned preference for tolerant clients.** Juvenile cleaners' proportion of choosing the plates that were tolerant (non-responsive to cleaners eating preferred prawn items) in the social learning phase of experiment 2. Dashed lines mark the preference score expected if juveniles chose at random between the tolerant and the responsive plate (0.5). Observer treatments (in grey): juveniles had witnessed adult demonstrators inducing one of the two plates to either flee or punish (chase the cleaner) in response to cleaners eating a preferred prawn item. Control treatments (in white): the same plates had been presented for similar amounts of time and removed in similar ways but without the demonstrators being able to eat off them (a transparent barrier prevented the demonstrator from approaching the plate, see Fig. 1). $N = 19$ individuals that participated in all treatments. Boxplots show the median and interquartile range, whiskers denote 1.5 × interquartile range and dots mark individual data points. Source data are provided as a Source Data file.

preferences. We allowed the juveniles to observe an adult repeatedly choosing between two simultaneously presented plate types that only differed in an arbitrary manner (their colour). The juveniles were divided into two treatments, whose goal was to generate variation in demonstrated preferences: in the 'preferring demonstrator' treatment, the demonstrators had been previously trained to clearly prefer one of the plates (plate role counterbalanced between observers). In the 'indifferent demonstrator' treatment they had been trained to approach the plates at random. Each observer could see the demonstrator's choices and the time it spent interacting with chosen plates (Fig. 4a). We then presented the observers with ten choice trials in which both plates were equally rewarding. If adults' preferences substantially influence the choices of juveniles, we would expect these preferences to be copied. Contrary to this expectation, juveniles' preferences were not affected by those of the demonstrators and did not differ between the two treatment groups (GLM: demonstrator preference: estimate $\pm$ SE $= -3.2368 \pm 3.019$, $N = 19$, $\chi^2 = 0.845$, d$f = 1$, $P = 0.358$; treatment group: estimate $\pm$ SE $= 0.817 \pm 1.056$, $N = 19$, $\chi^2 = 0.602$, d$f = 1$, $P = 0.438$, Fig. 4b). This suggests that although juvenile cleaners can use social information to learn about ecologically relevant cues pertaining to their complex social environment (as shown in our previous experiments), arbitrary effects induced by demonstrator's choices, or interaction time with a client, are unlikely to significantly affect their behaviour. Another result that points in this direction, is a finding in our first experiment, that demonstrators' feeding choices (number of flakes consumed) did not seem to affect the observers' subsequent levels of feeding against preference (LM: demonstrator's average flake consumption: estimate $\pm$ SE $= -0.115 \pm 0.226$, $N = 10$, $F = 0.261$, d$f = 1$, $P = 0.625$. Supplementary Fig. 1). Thus, it appears that the cleaners learned socially about the outcomes of the interaction rather than merely copying the demonstrators' behaviour.

## Discussion

Our results show the ability of a non-human animal to learn socially about behavioural strategies in an experimental paradigm that mimics cooperating/cheating in interspecific social interactions. Observation of adult cleaners' interactions facilitated the behavioural adjustment of juvenile cleaner fish: first, they socially learned to eat against preference in order to prolong interactions with model clients, an adjustment that would lead to behaving cooperatively in natural interactions with client fish. Second, they socially learned to strategically pick partners that yielded higher payoffs through the absence of evasive actions. Our results further suggest that information about the consequences of interactions was more salient to the juvenile cleaners than information about demonstrator's choices or feeding behaviour. This exhibition of payoff-based social learning in a social game, indicates that learning from observation can promote cooperative behaviour in self-serving ways.

Learning from observation about cooperation and cheating is often treated as a replication process, in which strategies are being copied by naïve individuals (e.g.,[5,7,20]). However, as the information encoded in cooperative interactions is multifaceted, social learning can be directed at different aspects of the interaction and elicit responses that do not necessarily match the behaviour being observed. In our experiments, the fish did not copy the demonstrators, but rather extracted information about the negative consequences associated with their behaviour. In experiments 1 and 3, the juveniles did not copy adults' behaviour or take their preferences into account, even when this was indeed feasible (for further discussion of the potential explanations for this result, see Supplementary Discussion section). In experiment 2, copying

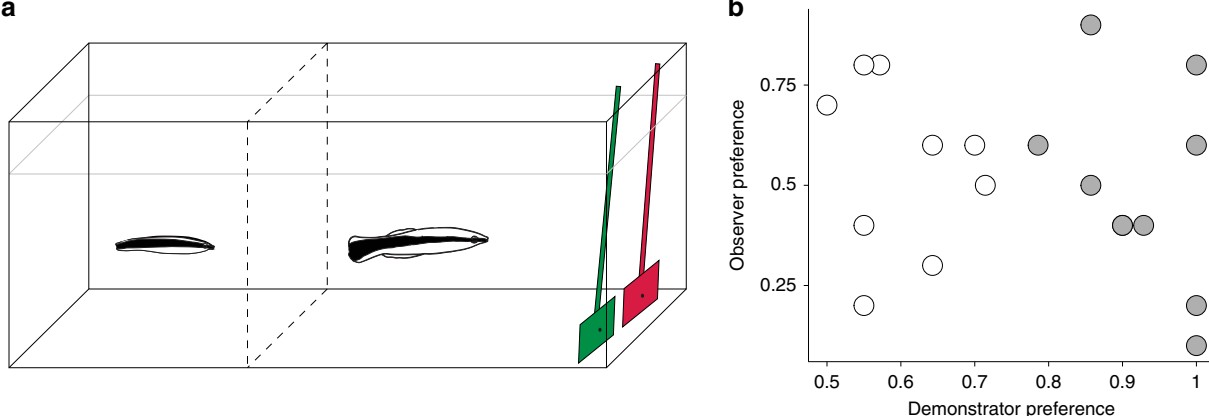

**Fig. 4 Observers do not copy arbitrary preferences.** Experiment 3, setup and results. **a** Demonstration phase setup. The compartments of juvenile and adult cleaners were separated via a transparent barrier, and in each demonstration session the juvenile cleaner (depicted on the left) could observe the adult making a choice between two differently colored plate types. The food was invisible to juveniles, located on the back of the plates (see black dots), and the plate not chosen was immediately retracted from the aquarium. **b** Plate preferences of juvenile observers during the test phase plotted against the preferences exhibited by their respective demonstrators in the training phase. Each dot depicts the proportion of choice of both the observer and demonstrator, for the plate that was more frequently chosen during demonstrations. Grey circles show the condition in which demonstrators had been trained to prefer one plate ($N = 10$), white circles show the condition in which demonstrators had been trained to approach the plates at random ($N = 9$). Source data are provided as a Source Data file.

was simply not possible—the juveniles did not see an adult choosing between a tolerant and an evasive plate, but rather, saw the consequences of interactions with both client types separately, and then had to make a choice based on this information.

In studies on the effects of social learning on cooperation in humans, payoff based social learning caused cooperation to deteriorate[22,23]. This may be expected in interactions with a prisoner's dilemma pay-off matrix, where cheating is dominant over cooperating. In our experiments, in which payoff based social learning promoted cooperation, behaving cooperatively towards model clients led to overall higher payoffs. Taken together, these results indicate that the effect of payoff based social learning on cooperation may not be exclusively detrimental. Instead, the payoff matrix underlying interactions is likely to dictate whether observation will lead to cooperation or to cheating. According to this logic, observation of cheating can induce cooperative behaviour if it leads to a more favourable outcome, which can occur when partner control mechanisms offset the benefits associated with cooperative and cheating strategies[2,3].

Can social learning about pay-off outcomes influence cooperation in other non-human species? The current lack of evidence for social learning of cooperative strategies[11] makes it hard to make concrete predictions. However, as animals learn socially in a variety of contexts, and can extract information from seeing other types of interactions[15,39,40] (see Supplementary Discussion for specific examples), we believe that social learning about cooperation is more prevalent than currently recognised. Nevertheless, social learning about effective cooperative or cheating strategies may also be constrained by difficulties associated with tracking payoff related information. Studies of payoff based social learning in non-cooperative contexts, indicate that success in learning about differential payoffs can be variable at both the inter- and intraspecific levels[41–44]. In experiments with chimpanzees and vervet monkeys, evidence for the use of social information for learning about more favourable outcomes is rather limited[42–44], while the evidence for nine-spined sticklebacks is strong[41]. Whether social learning about cooperation and cheating leads to higher payoffs, and the precise impact that this will have on interaction dynamics, will therefore depend on the content of the information that animals manage to extract from observing others. This can further differ between contexts and

individuals[45], as indicated by the variation in humans' use of social learning strategies in social dilemmas[21,46,47].

Theoretical models on reputational effects and their influence on cooperation have largely focused on how helping can induce higher cooperation levels[48–50]. However, negative reactions to cheating can also have reputational effects[51]. Such a reputation can promote cooperative behaviour in bystanders (as in the case of punishment in humans[52,53]), but may also bare negative consequences: in humans, bystanders reward helpers more than punishers[54,55] and punishers pay to hide their punishment from observers in lab-based economic games[56]. Here, we find that observation of plates responding negatively to the behaviour of adults affects the subsequent behaviour of juvenile cleaners towards these plates. Observation of plates' negative reactions incited the cleaners to behave more cooperatively (a response that would benefit real clients), but also to avoid responsive plates when alternative options were available (a response that would negatively affect real clients). It is hence plausible that real clients, as at least equally salient interaction partners, gain reputation based on their responses to cheating and that such reputation affects interaction dynamics in a natural model system. Using partner control mechanisms to respond negatively to cheating can thus act as a double-edged sword, favourably increasing cooperating in eavesdroppers, but reducing the chance of being chosen as interaction partner.

## Methods

**Subjects and housing**. This research was conducted in March–April 2018, at the Lizard Island Research Station, Australia. A total of 40 fish participated in the experiments, 20 juvenile and 20 adult cleaner wrasse, distinguishable by their distinct colour pattern: juveniles are mostly blue with a black stripe, while the blue becomes whiter in adults. There was no overlap between age classes with respect to body length (juvenile cleaners: average = 4.45 ± 0.64 SD cm; adult cleaners: average = 7.725 ± 0.56 SD cm). The cleaners were captured in the reefs surrounding Lizard Island at least 24 days prior to the beginning of the experiments and housed in the lab in separate aquaria with a constant flow of running seawater. Each fish was provided with a polyvinyl chloride tube for shelter. Before the experiments began, the cleaners were fed daily with mashed prawn that was smeared on plexiglass plates of varying colours and sizes. The fish were returned to their original habitat at the end of the experiments.

Three experiments were conducted consecutively, in two cohorts, each constituting 9–10 pairs of young and adult cleaners ($N = 20$ cleaner pairs in experiment 1, and $N = 19$ in the following experiments due to the death of a juvenile in the first cohort). Prior to the beginning of the experiments, each pair of

cleaners was placed in a glass aquarium, separated into two compartments by a clear partition. An additional opaque barrier was used to prevent visual contact between the cleaners during different stages of the experiments.

**Experimental setup.** In experiments 1 and 2, plexiglass plates were used as surrogates for clients, mucus was replaced with mashed prawn and ectoparasites with flakes. Both food types were used to create discrete food items that were placed on the plates. Flake items were made by mixing mashed prawn with commercial fish flake paste—mixing the prawn with this substance caused the food to be less palatable, and the higher the flake concentration in the mixture the less palatable it became. Levers attached to each plate allowed the experimenter to manipulate plates to 'behave' in pre-defined ways that mimic responses of real client fish. Plates never responded when a cleaner ate a non-preferred flake item, which corresponds to feeding cooperatively on non-preferred ectoparasites[28]. In contrast, experimenters made plates respond to a cleaner eating preferred prawn, as the consumption of preferred food (mucus) constitutes cheating in the wild[28]. In some cases, the response was to quickly remove the plate, the equivalent of a client fleeing and in other cases, the experimenter chased the cleaner with the plate, the equivalent of a client punishing[31].

In each experiment, any fish that would not receive food during the sessions (i.e., juvenile cleaners and control group adults with no access to plates in the demonstration phase), were fed ad libitum with smeared mashed prawn for 20 min prior to the beginning of the sessions. The cleaners in the two aquarium compartments were visually isolated from one another during this feeding.

**Experiment 1: socially learning to feed against preference.** In the first experiment, we tested whether juvenile cleaner fish can learn socially to eat against their preference following observation of the negative consequences associated with 'cheating', the consumption of preferred food. In this experiment, the fish were presented with plates (size 10 × 15 cm) offering three flake and three prawn items. To facilitate item recognition, prawn items were placed inside black circles while flake items were placed in black triangles. The locations of these marking, and their respective food items, were randomly switched between sessions in all experimental phases (by using ten different versions of the plate, each involving a different orientation of the markings).

Prior to the beginning of the experiment, we conducted separate pre-experimental training to cleaners of the two age classes. The opaque barriers were placed in the aquaria throughout this training, prohibiting visual contact between the cleaners. In this pre-experimental phase, demonstrator cleaners were trained to feed against preference to obtain more food (also see[31]). They participated in 6 sessions, each involving the presentation of a plexiglass plate containing 2 prawn and 12 flake items. In each trial, the consumption of a prawn item led to the immediate removal of the plate for 60 s, following which the plate would be reintroduced into the aquarium until the next prawn is consumed. The consumption of the second prawn item, then led to the removal of the plate from the aquarium until the next trial. In contrast, juvenile cleaners were accustomed to feeding on plates containing both food types that did not respond to prawn consumption. They were exposed to numerous presentations of plates containing seven flake and seven prawn items (initially, different flake concentrations were used, in order to find a concentration that the young cleaners indeed dislike) and were thus well familiar with both food types.

At the end of the acclimation phase, the juveniles' preferences towards prawn were measured and used to create for each fish an initial preference score. This was achieved by testing the fish in a series of three preference tests in which they were again presented with plates containing 7:7 flake and prawn items. In each of the tests, we quantified the amount of flake items eaten in the first seven choices (the point in which half of the plate is depleted). We then combined the results of the three tests to calculate for each fish a preference score: the proportion of prawn items eaten during the initial phases of the tests. Since juvenile cleaners' initial food preference is expected to substantially affect their feeding adjustment in the test phase of the experiment (the less they like flakes, the less likely they are to feed on them in the tests), we wanted to make sure that the two treatment groups do not differ in their preference for prawn. To achieve this, the cleaners' preferences in the initial preference tests were taken into account in their allocation to the two treatments. The cleaners within each cohort were ranked according to their initial preference score, and these ranks were then used to split the fish into the two treatments: first, we paired the two fish with the highest rank, and randomly assigned each of them to a different treatment group (allocation was determined via a lottery). We then moved to the fish with the highest ranks out of the remaining batch and repeated this procedure until all fish were allocated. Finally, we verified that the juveniles of the two treatment did not differ in their initial preference for prawn nor in their body size (Wilcoxon rank sum 2-tailed test, $N = 20$: initial preference, $W = 48$, $P = 0.909$; body size, $W = 52.5$, $P = 0.879$).

The flake concentration used in the experiment was different for juvenile subjects and adult demonstrators. In the case of juveniles, flake concentration was ~60% in the first cohort and ~40% in the second. These cohort differences stem from the fact that in the first cohort, some of the juveniles hardly consumed flakes in the test phase. As we were concerned that this would prevent any treatment differences from being pronounced (if the juveniles refuse to feed on flakes, a lack of variation in their response can mask any effect of observation), in the second

cohort we reduced the flake concentration to 40%. Due to this inherent difference between cohorts, we included 'flake palatability' (referring to the difference between the two cohorts) as a fixed factor in the statistical analysis of this experiment. The flake concentration of adults was lower (~20% flakes), due to initial tests revealing that juveniles are much more tolerant towards flakes than adults are. Using only 20% flakes for adult demonstrators ensured that they would indeed feed regularly on flake items prior to eating a prawn item during demonstrations, thereby potentially enabling juveniles to learn through observation about the consequences of eating flake items vs. prawn items.

The experiment was divided into two phases: a demonstration phase, and a test phase. During the demonstration phase, juvenile cleaners were provided with the opportunity to witness 14 demonstrations of plate presentations (see Fig. 1a). The plates were marked with an additional white stripe on their left side, to distinguish them from the plates used earlier, in the pre-experimental training phase. Each plate contained three flake and three prawn items, the location of which was randomly varied between trials. In the treatment group ($N = 10$), the consumption of a prawn item by the demonstrator led to the immediate removal of the plate from the aquarium. In the control group ($N = 10$), an additional transparent barrier prohibited the adult from interacting with the plate (Fig. 1b). The duration of plate presentation (the time the plates stayed in the aquarium) was matched between treatment and control groups: each control individual was paired to an individual from the observer treatment, and in each trial round, the control plate was removed at the same time as the experimental plate had been removed in response to the demonstrator eating a prawn item in the paired trial.

At the end of the demonstration phase, we conducted 15 test trials (5 on the day of demonstration, and 10 during the following day), in which the juveniles were presented with the same plates, and the plates followed the same response rule that was shown during demonstrations. In each of these tests, we measured the amount of flake items that were consumed by the cleaners prior to the consumption of prawn (the consumption of a food item was indicated by cleaners' attachment of the mouth to the plate at the item's location). This design enabled the juveniles to both acquire information by observing adults, and gain personal experience by interacting with the plates during the tests. This parallels natural conditions, where juveniles' use of social information likely involves getting direct feedback. It further fits the notion that social learning is a biasing of individual learning by social stimuli rather than a completely distinct process (also see refs. [14,57–62]). Note, however, that in itself, individual learning would not be able to account for differences between observer and control treatments. If anything, a strong effect of individual learning would be expected to diminish any potential differences between these treatments, as it allows individuals of the control groups to fill in any knowledge gaps created by the lack of exposure to social information. Thus, any significant effects of social observation on juvenile performance, would be pronounced despite the potential for individual learning, and not because of it.

At the end of the experiment, we calculated for each fish an eating against preference score. In order to achieve this, we first calculated the cleaner's predicted feeding score: the number of flake items that it is expected to target prior to the consumption of prawn, on plates containing three flake and three prawn items. We used a derivation of a formula developed by Gingins and Bshary[63] for the exact same purpose:

$$\text{Predicted feeding score} = \frac{3(1 - x)}{2x + 1}$$

The formula combines the initial feeding preferences of the fish and the probabilities of eating preferred and less preferred items in a set of sequential choices. The feeding preference score measured prior to the beginning of the experiment (proportion of prawn consumed in the tests—see details above) is denoted by $x$. The probability of eating a flake item is denoted by $1 - x$. This probability changes each time a flake item is consumed, as its consumption increases the proportion of prawn in the remaining items. The formula takes into account these changes in probabilities and generates for each fish a predicted feeding score: the number of flake items it is expected to eat prior to the consumption of prawn in a single test session. This score was then subtracted from the average number of flake items that the fish actually consumed in the tests (measured—predicted), thus generating a feeding against preference score. When the feeding against preference score is larger than 0, it means that the fish ate more flake items than predicted, whereas, when the feeding against preference score is lower than 0, the fish performed below the expected value.

**Experiment 2: socially learning about partner responsiveness.** In experiment 2, the juvenile cleaners ($N = 19$) participated in two consecutive experimental phases: an individual learning phase and a social learning phase, in which they were repeatedly confronted with the need to choose between plates that differ in their responsiveness to cleaners' consumption of preferred prawn. The two phases were comprised of several treatments: two in the individual learning phase, and four in the social learning phase (see further details in the description of each experimental phase below). Each of the fish participated in all treatments (a within subject design), and treatment order was counterbalanced between cleaners within each phase. Overall, during this experiment the juvenile cleaners encountered a total of twelve plates (size 7 × 9 cm) of different colours and patterns (see Supplementary Fig. 3). Four food items, two prawn items and two flake items, were placed on equally distant dots in the middle of each plate. Flake concentration was 40% for

the juvenile cleaners, and 20% in adult demonstrations. Food locations were counterbalanced, so that individuals were exposed to a variety of food location combinations throughout trials. One treatment was completed on each experimental day, and the roles assigned to the plates were counterbalanced between the treatments within each phase.

The first phase of the experiment tested for individual learning regarding plates' expected responses (IL, days 1–2). In this phase, the juvenile cleaners participated in two treatments, 'IL fleeing' and 'IL punishing', in which they could interact with the plates directly during the training phase while being visually isolated from the adults in the neighbouring compartments (Fig. 1c). Treatment order was counterbalanced between individuals. In each treatment, the cleaners were presented with plates belonging to two distinct types: a plate that allows the cleaner to eat what and as much as it wants ('tolerant to cheating'), gently leaving the aquarium when the cleaner is done feeding, and a plate that responds in a negative way if and when the cleaner eats a preferred prawn item ('responsive to cheating'). Plate responses varied between treatments: in the 'fleeing' treatment, the responsive plates would leave the aquarium abruptly following prawn consumption, whereas in the 'punishing' treatment, plates would chase the cleaner in the aquarium for ~3 s. Each treatment was comprised of 20 training sessions involving 10 single presentations of each plate type. Presentation order was determined by a lottery, with the constraint that the same plate would not be presented in more than three consecutive trials.

At the end of the training, the juveniles experienced a series of ten test trials in which they were allowed to choose between the two plates. Plates' reaction rules in these tests matched the rules they followed during training, and in each test trial, the plate not chosen was removed from the aquarium. Plate positions (left or right side in the aquarium) were counterbalanced between tests. On the first day of testing, the cleaners were relatively less active towards the end of the day, and the number of tests conducted was therefore reduced to eight. As the order of treatments was counterbalanced between individuals, this difference in the amount of testing between day 1 and day 2 could not have led to systematic biases that produce significant results. The results of this phase are reported in Supplementary Note 1.

The second phase of the experiment was a social learning phase (SL, days 3–6), in which the juvenile cleaners participated in four treatments involving different combinations of social observation and plate response type (a 2 × 2 matrix): observing adult demonstrators interacting vs. not interacting with plates, and responsive plate punishing or fleeing. Treatment order was counterbalanced between individuals. In each treatment, the juveniles were provided with the opportunity to witness 20 demonstrations, 10 of each plate type ('tolerant' or 'responsive'), in which the designated plate was placed in the demonstrator compartment. Observer cleaners could then observe the actual interaction of the adult with the plate (Fig. 1a). Control individuals could not witness an actual interaction as the adults were prevented from feeding on the plate by a transparent barrier (Fig. 1b). However, each control individual was paired to an individual from a respective treatment group. The order in which plates were presented, the time they spent in the aquarium and their departure rules (leaving gently, fleeing abruptly or 'punishing' by simulating the punishing movement pattern used in the observer treatment, but out of demonstrator's reach, within the area in which the plate is confined) were all matched to those used in the demonstrations of the paired observer. This allowed us to verify that any difference in preferences between control and observer treatments, will not be driven by differences in length of exposure to the two plates, or simply by their movement patterns. Due to the odd number of cleaners one individual was paired to itself: for this individual, each observation treatment was conducted prior to the relevant control, and both plate presentation order and exposure time used during the control treatments, were matched to the ones exhibited in the observer treatments.

At the end of the demonstration phase, an opaque barrier was placed in the aquarium, visually isolating the two compartments. The juvenile cleaners were then tested in a set of ten test trials in which they were allowed to choose between the two plates. Similarly to the individual learning phase, plates' reaction rules in these tests matched the rules they followed during training, and in each test trial, the plate not chosen was removed from the aquarium. Plate positions (left or right side in the aquarium) were counterbalanced between tests. This testing procedure enabled the juveniles to acquire personal experience in their interactions with the plates during the test phase. However, as explained in detail in the description of experiment 1, personal experience in itself would not be able to account for differences between observer and control treatments.

**Experiment 3: observing demonstrators' arbitrary preferences.** In experiment 3, the juvenile cleaners were allowed to observe demonstrations in which the adult would choose between two plates types. The juveniles' preference for either of the plates was then examined in a set of choice tests. The two plates used in this experiment were monochromatic plates (red and green, size 5 × 8 cm) with food items placed on dots drawn on their back. This ensured that the cleaners would not be able to see the food itself before making a choice, thus forcing them to base their choices on the colour cues rather than just readily approaching the visible food. Placing the food on the back of the plates further allowed us to pre-train the demonstrators prior to the beginning of the experiments, while using different reward regimes. Plate positions (left or right side in the aquarium) were counterbalanced in each of the experimental phases.

During pre-experimental training, the barrier between the demonstrator and observer compartments was opaque, preventing juvenile cleaners from observing adults' interactions with the plates. The juvenile cleaners were pre-trained to find food on the back of plexiglass plates by swimming behind the plate. The adult cleaners participated in at least 24 training sessions (some of them received more training, depending on their achievements, see further below), involving a choice between the two plate types. In each session, the two plates were placed in the aquarium, and following the adult's choice, the plate not chosen was immediately removed. The adults were divided into two treatment groups: in the 'preferring demonstrator' treatment, the demonstrators were trained to significantly prefer one of the plates (plate type counterbalanced between cleaners): in this treatment, the designated plate contained two prawn items and was thus always rewarding, whereas the other plate was empty of food. In the 'non-preferring' demonstrator treatment, the goal of the training was to cause the demonstrators to approach the plates in a random manner. The adults were thus trained with plates that were equally rewarding, both containing one prawn item.

In some cases, we provided the adults with additional pre-training sessions. In the 'non-preferring' treatment, this occurred when demonstrators showed a strong preference towards one of the plates (chose repeatedly the same plate type in three sequential trials). These demonstrators would then receive some trials in which the preferred plate did not offer any food to reduce the preference. In cases in which the demonstrators nevertheless exhibited difficulties in switching between the plates, the experimenter would allow the cleaner to explore both plates, rather than immediately removing the non-chosen (and rewarding) plate. In the 'preferring' treatment, additional trials were added if demonstrators showed weak preferences at the end of the pre-training (non-significant preference in the 24 sessions, according to a binomial test). This was conducted in order to maximise the chance that they will indeed show clear preferences in the experiment itself.

As in previous experiments, the experiment was divided into two phases: social observation and testing. In the social observation phase, the juveniles were provided with the opportunity to witness numerous demonstrations of plate presentations (14 in the first cohort and 20 in the second cohort). Food was placed on the back of the plates (and thus was not visible to the cleaners), and its allocation was similar to that of the adults' initial training period: 2 vs. 0 items in the preferring demonstrator treatment ($N = 10$), and 1 vs. 1 in the non-preferring treatment ($N = 9$). At the beginning of each trial, the demonstrator was confined to the side adjacent to the observer, and the two plates were placed in the aquarium. The barrier confining the demonstrator was then removed, allowing it to swim towards the plates. Following a choice of one of the plates, the plate not chosen was retracted. The remaining plate was removed from the aquarium only after the interaction ended and the cleaner swam away from it. This could take quite some time, as the cleaners would often provide the plates with tactile stimulation, a 'massaging' behaviour that is part of their service to clients in the reef[64]. Note that the demonstrators' choices in this phase, indicate that our initial training reached its goal, and generated variation in demonstrated preferences as well as clear differences in the demonstrations of the two treatment groups (Wilcoxon rank sum: $n = 19$, $W = 8.5$, $P = 0.003$. Compare the x-axis values of grey and white dots in Fig. 4b).

At the end of the demonstration phase, the opaque barrier separating the two compartments was introduced, and the juvenile cleaners were presented with a set of ten simultaneous choice tests between the same two plates. In these tests, one prawn item was located at the back of each plate, making the plates of equal value. As during demonstrations, the plate not chosen was removed from the aquarium before the juveniles could feed off it.

**Statistical analysis.** Statistical analyses were conducted in the statistical software R version 3.5.1[65]. The results of the different experiments were analysed using linear models, generalised linear models, and generalised linear mixed models, where applicable. Continuous predictors were all standardised using the function scale() from the base package in R language. Reported $P$ values were extracted using the function Anova() from R car package. Summary outcomes from all the fitted models are available in Supplementary Table 1.

In experiment 1, we were interested in whether juvenile cleaners adjusted the extent to which they ate against preference following observation. We tested the juveniles' feeding adjustment by fitting a linear model (model A, Supplementary Table 1) in which their calculated eating against preference score served as the response variable. Treatment group (observers vs. control) and flake palatability (the experiment was run in two cohorts, differing in the palatability of flake items) were added as fixed predictors. The formula syntax of this model was the following: eating against preference score ~ treatment group + flake palatability. We checked the model's diagnostics, normality of residuals and homogeneity of variances visually, by using residual plots and qqplots, and statistically, by using Shapiro–Wilk and Levene's tests.

In addition, we were also interested in whether the adults' feeding choices in the demonstration phase affected the feeding adjustment of juveniles in the observer treatment (control individuals did not observe the adults consuming food and were therefore not included in this analysis). To that aim, we fitted a linear model (model B, Supplementary Table 1) in which the juveniles' eating against preference score served as the response variable. Demonstrators' average flake consumption during observation and flake palatability served as predictors. The formula syntax

of this model was the following: eating against preference score ~ demonstrator's flake consumption + flake palatability. Tests of models' assumptions were similar to those described above for model A.

In experiment 2, we tested the individual learning and social learning phases separately ($N = 19$ in each phase), by fitting two generalised liner mixed models (GLMM with a binomial distribution, function glmer() from the lme4 package in R lunguage[66]). In the individual learning model, the response variable was the cleaners' binary choices between responsive and tolerant plates in the test phase, plate response type was fitted as a fixed predictor and cleaner identity as a random factor (See Supplementary Note 1). The formula syntax of this model was the following: cleaner's choice ~ plate response type + (cleaner identity as a random factor). In the social learning model (model C, Supplementary Table 1), cleaners' binary choices between responsive and tolerant plates was again the response variable. Social observation, response type and the interaction between them were fitted as fixed predictors, and cleaner identity was fitted as a random factor. The formula syntax of this model was cleaner's choice ~ social observation * plate response type + (cleaner identity as a random factor). For both models, we checked the models' diagnostics using residual plots, and the normality of residuals for the random factors using qqplots and Shapiro–Wilk tests. R squared for the mixed models was obtained using the function r.squareGLMM(), from the MuMin R package. In addition, we measured the performance of the fish in each treatment, by testing whether their preferences deviated from those expected by random choice (i.e., whether the proportion of tolerant plate choice in each treatment differed from 0.5). This was done using general linear hypotheses tests (GLHT) for the estimates of the mixed models, and $P$ values were adjusted using the Holm method.

In experiment 3, we tested whether treatment group and/or demonstrators' exhibited preferences affected the subsequent plate preferences of the observer juvenile cleaners. We fitted a generalized linear model (GLM) with a quasibinomial distribution; a distribution that controlled for the overdispersion of the residuals (by using an additional scale parameter). In this model, observers' choices of demonstrators' preferred and less preferred plates in the tests was the response variable, while demonstrators' preferences (proportion of choices of the more preferred plate of each demonstrator during the demonstration phase) and treatment group were the predictors. The formula syntax of this model was the following: cleaner's choices of demonstrator's preferred vs. less preferred plates ~ treatment group + demonstrator's preference. We checked the models' diagnostics (homogeneity of variance, residual normality and potential violations of linearity) using residual plots.

**Ethical note**. This study was performed in accordance with the guideline of the Animal ethics committee, Queensland, Australia (approval number: CA 2018/01/1156).

**Reporting summary**. Further information on research design is available in the Nature Research Reporting Summary linked to this article.

## Data availability

The data that support the findings of this study are deposited in Figshare data repository: https://doi.org/10.6084/m9.figshare.8068280.v1[67]. Source data for Figs. 2–4 and Supplementary Figs. 1 and 2 are provided in a Source Data file.

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

## Acknowledgements

We thank the friendly staff of Lizard Island Research Station for their help and support in the field; R. Slobodeanu for his help with statistical analysis; M. Burton-Chellew, O. Kolodny, A. Moran, and Y. Prat for helpful comments and discussion. This work was funded by SNF grant 310030B_173334 / 1 awarded to R.B., and a Swiss Government Excellence Scholarship (FCS) awarded to N.T.

## Author contributions

N.T. and R.B. designed the study. N.T. and Y.E. carried out data collection. N.T. analysed the data and led the writing. All authors discussed the results and commented on the paper at all stages.

## Competing interests

The authors declare no competing interests.
