## [Peer Review File · Nature Communications]

Reviewers' Comments:

Reviewer #1:

Remarks to the Author:

This paper shows that juvenile cleaner fish can learn about the consequences of interacting with particular clients by moderating their own behaviour to prolong a cooperative interactions. Interestingly, the young fish did not pay any attention to arbitrary client preferences (here based on colour).

The fact that fish can learn socially has been known for a very long time (see reviews by Brown and Laland), but here the experiments provide some rather interesting information on what the observer fish are learning having observed adults interact with clients that respond in various ways. The cooperation context is certainly novel.

It may well be the case that social learning is particularly important in these complex social interactions, wherein relying solely on individual learning may significantly reduce the probability on discovering the optimal social strategy. Perhaps more could be said about this in the discussion.

Given the widespread use of social learning in anti-predator contexts, where individuals rapidly learn the negative consequences of predation (punishment), the results of this experiment are perhaps less surprising, if not somewhat expected. Again this may be worth mentioning in the discussion.

Overall the paper is very well presented, the stats are appropriate and the results reasonably easy to follow.

Specific comments:

L48: delete "as"

L90: briefly say that younger observers were more inclined to adjust their preference.

L150: missing a full-stop at the end of the sentence

L213: insert a space before the reference

L346: replace "are" with "were"

Reviewer #2:

Remarks to the Author:

This paper reports a series of interesting findings on how cleaner fish can learn by observational conditioning what type of response to expect from arbitrarily marked plates, intended to represent simulated cleaner clients. I find the results overall convincing although I have a number of substantive questions about the statistical analysis, which seems oddly convoluted given that with a designed experiment like this it should have been possible (and desirable) to specify and use a single model incorporating the logic of the experimental design to specify the effects of interest and any additional confounds resulting from the design. I also think that the authors have been somewhat uncritical in directly comparing their results using simulated plates with results from humans in actually-interacting experimental settings. I can buy that cleaners can learn by observational conditioning the properties and behaviour of specific objects in their environment, but I am not so convinced this can be directly translated into the natural setting - to illustrate this, imagine the counter-factual experiment on humans where instead of other humans the subjects observed interactions with objects - would we really be happy describing this as learning about 'reputation' as opposed to learning about the properties of objects in the environment?

Specific comments:

L22-24 'They further show that negative responses to cheating can shape the reputation of cheated individuals' - the 'show' in the abstract here is not consistent with the 'plausible' when this is addressed in the discussion L165: 'It is hence plausible that real clients, as at least equally salient interaction partners, gain reputation' - this is unfortunate, and the abstract should be revised.

There are a number of grammatical errors that should be corrected e.g. L47 (and elsewhere) 'which constitutes as cheating' L194 'clients where represented by' L345 tense agreement L359 redundant comma

L86 It would be preferable to present the complete model summary in a table I feel so that the reader can see all parameters estimated and not just the reported ones - ideally this should also include the full (global) model (see below).

L105 'the effect of observation was significant, although diminishing' - I don't understand the 'diminishing' part here?

L191-193 direct repeat

L195-196 - 'flakes' contain prawn also?

I got confused over '% flake concentration' vs number of flake patches presented perhaps this can be clarified

L272 "Plates' reaction rules in these tests matched the rules they followed during training, and in each test trial..." - can the authors be confident, and thus assert, that in assigning plate patterns to specific behaviour patterns they were not inadvertently matching reaction styles to patterns and/or colours that might be expected to have specific ecological relevance and therefore provoke innate responses (e.g. for example the colour red can do this in some species?)

L293 in the control condition what was the cue from the demonstrator that initiated plate 'behaviour', and how could the chasing punishment be demonstrated when there was a transparent barrier between the plate and fish?

L344 'The significance of dropped predictors was obtained by adding them to this minimal model.' - I am confused about why this was done? The authors should note that inferential tests following data dependent model selection are unlikely to be valid (see e.g. Heinze et al 2018 doi: 10.1002/bimj.201700067). It's unclear to me why any model selection was needed at all, given that the experiment design was known a priori it should be possible to specify a model in advance and the most appropriate one would appear prima facie to be the full (global) one.. In any case, it would be good practise for the authors to report these global models, at least in the SI. Finally, it is important if AICc is being used that the deltaAICcs are reported between the 'best' and other models - if there is model selection uncertainty (i.e. $\Delta < 2$), then the authors need to report this and account for it...

L350-351 - "we centred the cleaners' initial preference for prawn according to the cohort in which they participated" - this is not well justified for me - I can understand centering on each individual's initial preference so that preference at test is presented relative to preference pre-demonstration, but it is unclear why the behaviour of different fish that have never interacted with the focal should have any impact on the representation of the focal fish's behaviour - if the authors suspect cohort specific

effects then isn't it most appropriate to model these explicitly using a multi-level approach (as for example in experiment 2)?

L351 - averaged across what? Why should this be an average, and not an explicit model of the individual responses?

L355 - "The non-linearity" - what non-linearity? I can't find any other mention of this?

L360 'we fitted a linear model between' - please specify which are the dependent and independent variables here. Also justify why this needs to be a separate model rather than including the demonstrator consumption in the global model?

L376 'The statistical significance of cleaners' preferences was obtained from simpler versions of these models, taking only the treatment groups into account' - I cannot understand how this can be correct - such models will ignore any confounding effects of the various other predictors used in the global model - surely the appropriate statistical test is from the full (global) model?

L381 justify choice of quasibinomial model here.

L386-388 - there is an inconsistency here with the model selection approach outlined earlier - why is it relevant that nothing was significant in the full model?

Fig 2/3 it would be very helpful for rapid comprehension to have a visual legend on these figure. I would also like to see the individual data points added to these figures to understand just how much the outlier in the cohort 1 observer treatment is driving the 'average' response.

Fig 3 does the different shading indicate anything additional over what is already indicated by the group labels - if not, it is redundant and potentially confusing.

Fig 4 caption not always grammatical

SI "However, as it seemed to us that for some of the cleaners, this concentration was very strong..."

The following is unclear to me: "To make sure that our treatment groups would not differ in their initial preference, within each cohort, we ranked the cleaners' preferences, split them into pairs based on these ranks, and randomly allocated the two cleaners in each pair to the two different treatments"

What were the demonstrator training performance criteria? " Demonstrators that showed weak preferences at the end of training (non-significant according to a two-tailed Wilcoxon test), would receive additional training trials. "

Reviewer #3:

Remarks to the Author:

This is a very interesting paper building on a sound body of prior work with cooperation in cleaner fish. The investigation of social learning's role in cooperative interactions is long over due in animals and the paper will be of interest to many researchers. Detailed comments below, but my main comments are that:

(1) In places, the literature referred to in the Introduction and used in the Discussion needs

broadening.

(2) The Results section is often not sufficiently explanatory of the methods for the reader to understand and interpret the validity of the results reported.

(3) There is no consideration that repeated test trials allowed individual learning (that coincided with the social information) for some of the observer fish. Statistics reporting the results of the first test trials (after observation/non-observation phase) are required.

(4) The findings may be a little underplayed

a) it is unusual in the social learning literature to find an example of social learning NOT to perform a behaviour pattern.

b) more could be made of the finding that the fish only socially learned behaviours that had tangible outcomes (although this statement would need validation with other basic arbitrary preference tests)

(5) The Discussion does not really discuss any finer nuances of the study. For example, what is the ecological validity of the various experiments and do the experiments leave any unanswered questions or require replication? etc.

Rachel Kendal & PhD student

Detailed comments

Abstract

Line 15 – if you just mean ‘unlearned’ use that instead of ‘instinct’ as the latter has many interpretations.

Line 19: socially learning not to perform a behaviour is something not reported often, more could be made of this

Introduction

Line 33 – Also see Richerson et, al., (2016) “Cultural group selection plays an essential role in explaining human cooperation: A sketch of the evidence” for a further review into this topic.

Line 35 – Consider rephrasing this statement as it is not fully clear at first glance which “phenomena” are being referred to.

Line 40 – May be worth briefly mentioning social learning strategies and the individual differences / cultural variation present in social learning (specifically for social dilemmas). For example: Molleman, L., Van den Berg, P., & Weissing, F. J. (2014). Consistent individual differences in human social learning strategies. *Nature Communications*, 5, 3570.

Molleman, L., & Gächter, S. (2018). Societal background influences social learning in cooperative decision making. *Evolution and Human Behavior*, 39(5), 547-555.

Lamba, S. (2014). Social learning in cooperative dilemmas. *Proceedings of the Royal Society B: Biological Sciences*, 281(1787), 20140417.

Replace the term ‘conveyed’ with ‘used’.

Line 42 – For clarity, perhaps explain how social learning to adjust cooperation could be beneficial first before moving on to relate it to cleaner fish.

Line 50 – Do cleaner fish then respond to this by increasing their cooperation with that individual? Or do they permanently terminate their interaction?

Line 55 – you can refer to the social learning strategies/transmission bias literature to validate the assumptions of social learning in juvenile fish (ie) copy when ignorant, copy when asocial learning is costly etc. see Kendal RL, Boogert NJ, Rendell L, Laland KN, Webster M & Jones PL. (2018). Social Learning Strategies: Bridge-building between fields. *Trends in Cognitive Sciences* 22(7): 651-665.

Line 72 – it would be useful here to explain briefly what the ‘evasive response’ actually was (ie) how did the plate move (where to?), did it ‘chase’ the cleaner fish?

It would be useful to give the rationale for the three different experiments before moving onto the results.

Results

Line 80 – As described here the control does not sound like a true control? Everything was not held constant except the adult interaction with the plate (ie) the plate/client seems not to move in the

control. However, on line 234-6 we discover that the control was very good. Please give this information here.

Line 87-91 – The explanation of the results here is not clear. How were their preferences influenced? What is flake concentration? What is 'feeding adjustment'?

Line 98 – not clear how this makes 4 treatments, rather than 6?

Line 99 – how could observer fish distinguish between tolerant and responsive plates? Were they different colours/sizes/shapes? Was the order in which fish experienced treatments counterbalanced? If not does this invalidate the results?

Line 103 – was there not also a control where the plate did not move (ie) control for 'tolerant' plate.

Line 107 – confusingly written "only cleaners of the observer treatments..." Do you mean only following 'observer trials' as each fish received each of the 4 treatments (according to line 98)?

Line 115 – state what you actually mean by 'blindly'? Do you just mean copying of an arbitrary preference?

Line 116 – was this a between subjects design?

Line 119 – was the preferred plate counterbalanced across observers?

Line 124 – Consider rephrasing for clarity. Perhaps just "We would expect these preferences to be copied".

Line 135 – The way this sentence is framed slightly implies that the fish learned about outcomes instead of copying behaviour patterns. This and the proceeding discussion could be rephrased to more clearly emphasise that cleaner fish seemed selective in which behaviours they copied. They only copied behaviour patterns that resulted in tangible outcomes, which is interesting but underplayed.
Discussion

Line 145 – but were they actually given a choice of plates at test? This needs clarifying in the results section.

Line 149 – the authors should consider and compare with other literature that has considered payoff based social learning in animals eg.

Vale, G, Flynn, EG., Kendal, JR., Rawlings, B, Hopper LM., Schapiro SJ., Lambeth SP. & Kendal RL. (2017). Testing differential use of payoff-biased social learning strategies in children and chimpanzees. *Proceedings of the Royal Society B*

Bono AEJ, Whiten A., van Schaik C., Krützen M., Eichenberger F., Schneider A., van de Waal E., 2018. Payoff- and Sex-Biased Social Learning Interact in a Wild Primate Population. *Current Biology*, 28 (17) pp. 2800-2805.e4. Peer-reviewed.

Line 155 – It would be helpful to consider specific examples. The major alternative which comes to mind is the Snowdrift game.

Doebeli, M., & Hauert, C. (2005). Models of cooperation based on the Prisoner's Dilemma and the Snowdrift game. *Ecology letters*, 8(7), 748-766.

Line 159 – Consider using a simpler word rather than "hitherto". Perhaps just "currently".

Line 165 – "have an equivalent effect AS CHASERS on juveniles"?

Line 174-5 - This could be clearer. Do you mean punishment increases cooperation in eavesdroppers but reduces the chance that they choose to interact with the punisher they observed? Please clarify
In general there is no discussion of ecological validity of the studies. For example, could an explanation for the lack of social learning of an apparent arbitrary preference for one coloured plate over another (experiment 3) be explained by it not representing the differences between clients effectively? What cues do cleaner fish use to distinguish clients, perhaps size or pattern is important and this did not vary between the two plates.

Materials and Methods

Line 191-194 – unnecessary repetition

Line 213 – Typo, missing space (or missing reference) in brackets.

Line 214 – who/what "didn't respond to prawn consumption"?

Line 238 – the repeated 15 trials at test (with the plates responding as during observation phase)

indicate that individual learning could easily have taken place. What were the results of the very first trial? This is the one for which no individual learning could have yet obscured results? Is there the same problem with experiment 2 (Line 298)?

Line 241: what qualified as a 'flake item' being consumed? Did the fish have to nibble at it or consume it entirely for it to be counted as eaten?

Line 245: experiment 2 is described as having 4 treatments in the Results (this needs clarifying).

Line 322 – Why is it beneficial for the prawns to be on the back of the plates?

Statistical Analysis

Line 344 – For those unfamiliar, it would be useful to briefly explain AIC and how it is used to evaluate models.

Line 381 – May be useful to explain the quasibinomial distribution and how it is different to a regular binomial distribution.

Reviewers' comments:

Reviewer #1 (Remarks to the Author):

This paper shows that juvenile cleaner fish can learn about the consequences of interacting with particular clients by moderating their own behaviour to prolong a cooperative interactions. Interestingly, the young fish did not pay any attention to arbitrary client preferences (here based on colour).

The fact that fish can learn socially has been known for a very long time (see reviews by Brown and Laland), but here the experiments provide some rather interesting information on what the observer fish are learning having observed adults interact with clients that respond in various ways. The cooperation context is certainly novel.

It may well be the case that social learning is particularly important in these complex social interactions, wherein relying solely on individual learning may significantly reduce the probability on discovering the optimal social strategy. Perhaps more could be said about this in the discussion.

Given the widespread use of social learning in anti-predator contexts, where individuals rapidly learn the negative consequences of predation (punishment), the results of this experiment are perhaps less surprising, if not somewhat expected. Again this may be worth mentioning in the discussion.

Overall the paper is very well presented, the stats are appropriate and the results reasonably easy to follow.

- Thank you - this was nice to hear.
We added further details regarding the potential relevance of social learning to animals interacting in complex social environments in general and juvenile cleaners in particular (see introduction: L 53-82, in addition to 92-98). We certainly agree with the reviewer that animals can learn about negative consequences in other contexts. We now address this in the main text (L 324-326), and thoroughly discuss it in the supplementary discussion (page 6). Nevertheless, note that at least in the case of our first experiment, the cleaners learned socially to inhibit their choices of preferred food, a challenge that we believe is more complex.

Specific comments:

L48: delete "as"

L150: missing a full-stop at the end of the sentence

L213: insert a space before the reference

- Thank you for pointing out different typos and grammatical errors – all were addressed in the text (L 66, 217, 320).

L90: briefly say that younger observers were more inclined to adjust their preference.

- Following the comments of reviewer #2, we modified and simplified our analysis. In order to not overfit the models, the analysis now only focuses on variables that were inherent to our experimental design (explicit experimental modifications), and cleaner size was excluded.

L346: replace “are” with “were”

- As the analysis was modified, this sentence was removed.

Reviewer #2 (Remarks to the Author):

This paper reports a series of interesting findings on how cleaner fish can learn by observational conditioning what type of response to expect from arbitrarily marked plates, intended to represent simulated cleaner clients. I find the results overall convincing although I have a number of substantive questions about the statistical analysis, which seems oddly convoluted given that with a designed experiment like this it should have been possible (and desirable) to specify and use a single model incorporating the logic of the experimental design to specify the effects of interest and any additional confounds resulting from the design.

- Thank you for this comment – we appreciate the feedback and agree that our original analysis was overly complex. We revised and substantially simplified the statistical analysis to better fit the main research hypotheses, removing variables that were not an inherent part of our experimental design (see further details in our responses to the specific comments below).

I also think that the authors have been somewhat uncritical in directly comparing their results using simulated plates with results from humans in actually-interacting experimental settings. I can buy that cleaners can learn by observational conditioning the properties and behaviour of specific objects in their environment, but I am not so convinced this can be directly translated into the natural setting - to illustrate this, imagine the counter-factual experiment on humans where instead of other humans the subjects observed interactions with objects - would we really be happy describing this as learning about 'reputation' as opposed to learning about the properties of objects in the environment?

- We tried to be careful with our interpretation, and highlight repeatedly that the paradigm involves simulated social interactions. Perhaps due to this approach, reviewer #3 even thought that we undersell the results. Specifically, in our discussion of the implications of our results for client reputation, we stress that such reputational effects are plausible (in the main text) or potential (in the modified abstract). We thank the reviewer for pointing out specific cases in which further caution could benefit the writing – we adjusted the text following these comments and think that these modifications indeed helped to improve the paper.
- Nevertheless, we wish to point out that the experimental paradigm we used in this study is well-established and simulates the different components of the interactions of cleaner fish with clients (e.g. Bshary & Grutter 2002, Grutter & Bshary 2003, Bshary & Grutter 2006, Raihani et al. 2011). Plates are not passive objects but act like agents do. In the past, attempts to replicate findings obtained using this paradigm in experiments with real client fish, were successful and showed similar patterns to experiments with model clients (Pinto et al. 2011). Although this paradigm cannot show directly the effect of

information transmission in the natural environment, it is very useful in showing that young cleaner fish are indeed capable of learning socially about the meaningful components of interactions that they are likely to observe in nature, and rely on such information to modify their own behavior.

As for social learning and cooperation in humans – although we appreciate the reviewer's concern, we do wish to point out that many of the experiments addressing both cooperation and social learning in humans involve games conducted on computer screens, one-shot interactions in which the participants do not see or know their partners, and in which social information is provided through written messages informing the focal individual regarding payoffs/investments of strangers (e.g. Molleman et al, 2014, van den Berg et al. 2015, Burton-Chellew et al. 2017). In addition, studies of different aspects of children's' behavior towards cooperation partners, often use puppets, and observation of their “behavior” towards others rather than real-life interaction partners (e.g. Hamlin et al. 2011, Riedl et al. 2015). Studies using these methods can nicely identify expected strategic behavior and modes of information transmission in different cooperation scenarios and illuminate potential influences of social learning on cooperation. However, we are not sure that we find these experimental settings more convincing in simulating actual cooperative interactions than our model client paradigm.

Specific comments:

L22-24 'They further show that negative responses to cheating can shape the reputation of cheated individuals' - the 'show' in the abstract here is not consistent with the 'plausible' when this is addressed in the discussion L165: 'It is hence plausible that real clients, as at least equally salient interaction partners, gain reputation' - this is unfortunate, and the abstract should be revised.

- This is a valid point. We adjusted the sentence in the abstract, and it now states that our results show that such responses can potentially shape reputation (L 23).

There are a number of grammatical errors that should be corrected e.g. L47 (and elsewhere) 'which constitutes as cheating' L194 'clients where represented by'

- Thank you for pointing out these errors – we corrected them in the text (L 66, 87, 303; 293)

L345 tense agreement L359 redundant comma

- These sentences were removed from the manuscript.

L86 It would be preferably to present the complete model summary in a table I feel so that the reader can see all parameters estimated and not jsut the reported ones - ideally this should also include the full (global) model (see below).

- In addition to reporting the results in the main text, we added a table with the summaries of the different models to the Supplementary information (page 3, Supplementary table 1). We no longer conduct model selection, and the tables reported show the results of full models.

L105 'he effect of observation was significant, although diminishing' - I don't understand the 'diminishing' part here?

- We initially included in our analysis a predictor that related to the order in which the treatments were conducted. After giving this additional thought, we realized that this predictor should not be included (since each cleaner experienced a unique treatment sequence, this would not allow us to make inferences regarding the effect of the order). This sentence was therefore removed.

L191-193 direct repeat

- This was indeed redundant! We removed one of the repetitions.

*L195-196 - 'flakes' contain prawn also?
I got confused over '%flake concentration' vs number of flake patches presented
perhaps this can be clarified*

- Thank you for bringing this to our attention. We added clarifications in the text (lines 294-298) and hope that it is now clearer.

L272 "Plates' reaction rules in these tests matched the rules they followed during training, and in each test trial..." - can the authors be confident, and thus assert, that in assigning plate patterns to specific behaviour patterns they were not inadvertently matching reaction styles to patterns and/or colours that might be expected to have specific ecological relevance and therefore provoke innate responses (e.g. for example the colour red can do this in some species?)

- Yes, we are certain that this was not the case, since the role that each specific plate played in our experiment, was counterbalanced (L 382-383). While biases resulting from innate preferences could potentially influence the cleaners' behavior, our design ensured that they would not be able to affect our results in a systematic manner. If such biases would indeed be strong, they would be expected to reduce differences between treatment and control groups rather than enhancing them.

L293 in the control condition what was the cue from the demonstrator that initiated plate 'behaviour', and how could the chasing punishment be demonstrated when there was a transparent barrier between the plate and fish?

- In the control treatment, the plate was presented in the aquarium, but separated from the demonstrator via an additional barrier. Its departure was not

contingent on the demonstrator's behavior, but dependent on the time it remained in the aquarium during the demonstration of the paired observer cleaner. Simulated punishment involved the movement pattern used during punishment, but within the compartment in which the plate was placed, and thus out of the demonstrator's reach. See clarification in lines (419-429).

L344 'The significance of dropped predictors was obtained by adding them to this minimal model.' - I am confused about why this was done? The authors should note that inferential tests following data dependent model selection are unlikely to be valid (see e.g. Heinze et al 2018 doi: 10.1002/bimj.201700067). It's unclear to me why any model selection was needed at all, given that the experiment design was known a priori it should be possible to specify a model in advance and the most appropriate one would appear prima facie to be the full (global) one.. In any case, it would be good practise for the authors to report these global models, at least in the SI. Finally, it is important if AICc is being used that the deltaAICcs are reported between the 'best' and other models - if there is model selection uncertainty (i.e. $\Delta < 2$), then the authors need to report this and account for it...

- Thank you for this comment, and for referring us to the Heinze et al. paper which we found helpful. Our initial analysis of the data of the different experiments included many variables, some of which were not really necessary for testing the main hypotheses. Following the reviewer's comment, we substantially revised the statistical analysis, constructing for each experiment a model that captures the logic of the experimental design and includes only variables that were an inherent part of it. In addition to reporting the results in the main text of the results section, we added to the SI a table that includes the summaries of the full models.

L350-351 - " we centred the cleaners' initial preference for prawn according to the cohort in which they participated" - this is not well justified for me - I can understand centering on each individuals initial preference so that preference at test is presented relative to preference pre-demonstration, but it is unclear why the behaviour of different fish that have never interacted with the focal should have any impact on the representation of the focal fish's behaviour - if the authors suspect cohort specific effects then isn't it most appropriate to model these explicitly using a multi-level approach (as for example in experiment 2)?

- Originally, we used this approach to further account for cohort differences in initial preference. We modified the analysis, and no longer use this centering procedure. Instead, we include flake palatability (i.e. cohort) in the models of this experiment (L 510-512, 523).

L351 - averaged across what? Why should this be an average, and not an explicit model of the individual responses?

- Each individual participated in 15 consecutive tests, and the measure initially used in our analysis was, for each individual, the average number of flakes it consumed in the tests. In the revised version of the manuscript, we modified

this analysis. For each individual we calculate an eating against preference measure (predicted performance based on initial preferences, subtracted from the actual average performance of each fish in the test phase). This measure is then used as the dependent variable in the linear model (L 500-510).

L355 - "The non-linearity" - what non-linearity? I can't find any other mention of this?

- As cleaner size was not part of our experimental manipulation, we no longer include it in our models. This sentence was therefore removed.

L360 'we fitted a linear model between' - please specify which are the dependent and independent variables here. Also justify why this needs to be a separate model rather than including the demonstrator consumption in the global model?

- The consumption of flake items by the demonstrator could not be included in our basic model since this variable is only available for individuals of the observer group. In the control treatment, the demonstrator did not interact with the plates, and therefore did not consume any food items during demonstrations. As we were nevertheless interested in the effect of demonstrators' food consumption on the choices of juveniles that observed it, we tested its effect in a separate model (the text was modified following the reviewer's request, see L 518-523).

L376 'The statistical significance of cleaners' preferences was obtained from simpler versions of these models, taking only the treatment groups into account' - I cannot understand how this can be correct - such models will ignore any confounding effects of the various other predictors used in the global model - surely the appropriate statistical test is from the full (global) model?

- Thank you for this comment – we corrected the explanation of this part. The statistical significance of cleaners' preferences was obtained from the main model of each experiment, by using a GLHT procedure (to compare the actual performance of each treatment to 0.5). Thus, no variables are ignored or precluded in this analysis, and it is derived directly from the estimates of the mixed model of each experiment. P value adjustment was further conducted. (L 544-548).

L381 justify choice of quasibinomial model here.

- We used a quasibinomial rather than a binomial distribution because the data was overdispersed. We now mention this explicitly in the text (L 552-553).

L386-388 - there is an inconsistency here with the model selection approach outlined earlier - why is it relevant that nothing was significant in the full model?

- Our modeling approach was substantially modified (see further above), and this sentence was removed from the text.

Fig 2/3 it would be very helpful for rapid comprehension to have a visual legend on these figure. I would also like to see the individual data points added to these figures to understand just how much the in the cohort 1 observer treatment is driving the 'average' response.

- We modified figure 1, and it now shows an eating against preference score rather than the number of flakes consumed. Both plots now also show the individual data points, and the subheadings of the X axes were modified, to further promote comprehension.

Fig 3 does the different shading indicate anything additional over what is already indicated by the group labels - if not, it is redundant and potentially confusing.

- It helps to distinguish between social observation and control treatments. As it is visually consistent with figure 2, we find it quite useful.

Fig 4 caption not always grammatical

- Thank you for drawing our attention to this, we modified the caption. (L 850-862)

SI "However, as it seemed to us that for some of the cleaners, this concentration was very strong..."

- In the first cohort, the flake concentration chosen was very high, and as a result, some of the juveniles hardly consumed flakes in the test phase. This could potentially prevent any treatment differences from being pronounced (if the juveniles refuse to feed on flakes, a lack of variation in their response can mask any effect of observation). As we were concerned by this possibility, we reduced the flake concentration in the second cohort (we modified the text in page 10 of the Supplementary Methods to make this clearer). This adjustment is represented by the flake palatability variable in the statistical analysis.

The following is unclear to me: "To make sure that our treatment groups would not differ in their initial preference, within each cohort, we ranked the cleaners' preferences, split them into pairs based on these ranks, and randomly allocated the two cleaners in each pair to the two different treatments"

- Since initial preferences for the two food types are expected to substantially affect juveniles' feeding adjustment in the test phase (the less they like flakes, the less likely they are to feed on them in the tests), we wanted to make sure that the two treatment groups do not differ in their initial preference for prawn. We therefore took the initial preference into account in our allocation of the juveniles to the two treatment groups. This was achieved by measuring the juveniles baseline preference via preference tests (As explained further in the text) and ranking them according to their score in these tests. The ranks were then used to split the fish into the two treatments: within each cohort, we first took the two fish that reached the highest ranks (i.e. obtained the strongest preference for prawn relative to the other fish in their cohort), and assigned

each of them to a different treatment group in a randomized manner (allocation was determined via a lottery). We then moved to the fish with the highest ranks out of the remaining batch and repeated this procedure until all fish were allocated. This procedure is now more thoroughly explained in the Supplementary Methods (page 10, L210-221).

What were the demonstrator training performance criteria? " Demonstrators that showed weak preferences at the end of training (non-significant according to a two-tailed Wilcoxon test), would receive additional training trials. "

- All demonstrators initially received 24 training trials. Since in the "preferring demonstrator" treatment, we wanted to maximize the chance that they will indeed show clear preferences in the demonstration phase, we added additional session in cases in which the demonstrators' preference was not pronounced at the end of these 24 trials. To define which demonstrators would receive additional training, we tested the overall preference of each of these cleaners at this point in time, via a binomial test. Demonstrators whose preference was not statistically significant according to this test received additional training. This point was further clarified in the Supplementary Methods (page 12, L266-269).

Reviewer #3 (Remarks to the Author):

This is a very interesting paper building on a sound body of prior work with cooperation in cleaner fish. The investigation of social learning's role in cooperative interactions is long over due in animals and the paper will be of interest to many researchers.

- We were very happy to hear that you found our paper to be interesting and relevant, and certainly agree that this topic is currently understudied in non-human animals.

Detailed comments below, but my main comments are that:

(1) In places, the literature referred to in the Introduction and used in the Discussion needs broadening.

- Thank you for making specific suggestions in your comments further below – they helped us to expand the discussion and better relate it to the existing literature. The discussion now addresses more topics and broader literature is cited in both introduction and discussion sections.

(2) The Results section is often not sufficiently explanatory of the methods for the reader to understand and interpret the validity of the results reported.

- We appreciate the feedback. As the design of some of the experiments is quite complex, it was not always easy to describe the results in a concise manner. We tried to improve this section following the comments and hope that it is now clearer.

(3) There is no consideration that repeated test trials allowed individual learning (that coincided with the social information) for some of the observer fish. Statistics reporting the results of the first test trials (after observation/non-observation phase) are required.

- This issue is now addressed in detail (see our response to a similar comment below).

(4) The findings may be a little underplayed

a) it is unusual in the social learning literature to find an example of social learning NOT to perform a behaviour pattern.

b) more could be made of the finding that the fish only socially learned behaviours that had tangible outcomes (although this statement would need validation with other basic arbitrary preference tests)

- Thank you for this comment. It was important to us to not oversell our findings, and we therefore tried to be very careful in our interpretation. We hope that you will find that the discussion of the results in the revised version better fits the scope of the findings.
We agree that the fact that juveniles did not copy adults can be made more explicit, and now further address this point in the discussion (L218-230). We further agree that learning seems to be selective, but also think that there are different interpretations to this result that cannot be ruled out. We now further discuss this. Due to space limitations (adding points to the discussion made it quite long), this point is addressed in the Supplementary Discussion (pages 6-7 L109-131).

(5) The Discussion does not really discuss any finer nuances of the study. For example, what is the ecological validity of the various experiments and do the experiments leave any unanswered questions or require replication? etc.

- As we tried to emphasize throughout the manuscript, our model client paradigm simulates key features of cleaner-client interactions, and the experimental design was tightly linked to cleaner fish ecology (we stress this explicitly at the introduction, and at the end of the different results sections L 83-95, 123-124, 134-136, 171-173, 193-197). We now further elaborate on specific nuances of the experimental design in the supplementary Discussion (pages 6-7).

Rachel Kendal & PhD student

Detailed comments

Abstract

Line 15 – if you just mean ‘unlearned’ use that instead of ‘instinct’ as the latter has many interpretations.

- We modified the sentence to “cooperation in other species might be restricted because it mostly relies on individually learned or innate behaviours” (L 15).

Line 19: socially learning not to perform a behaviour is something not reported often, more could be made of this

- Do you refer to the fact that the cleaners did not copy preferences? While we are aware of this trend, there are indeed examples in the literature of cases in which animals do not learn from observation (e.g. do not copy observed preferences) or learn about consequences rather than copying the behavior itself. With respect to the former, we further suspect that this is even more prevalent than it might seem, as publication biases are likely to favor positive results. Nevertheless, throughout the text, we tried to emphasize that the cleaners in our study seem to be learning from consequences rather than copying, and now further elaborate on this point in the discussion (L218-230) and the Supplementary Discussion (pages 6-7, L109-131).

Introduction

Line 33 – Also see Richerson et al., (2016) “Cultural group selection plays an essential role in explaining human cooperation: A sketch of the evidence” for a further review into this topic.

- The reference list was expanded, and this paper is now cited in the text.

Line 35 – Consider rephrasing this statement as it is not fully clear at first glance which “phenomena” are being referred to

- We rephrased the sentence, it now states that “Although both cooperation and social learning are widespread in nature, evidence for the use of social learning by non-human animals in cooperative contexts is extremely limited” (L 34-36).

Line 40 – May be worth briefly mentioning social learning strategies and the individual differences / cultural variation present in social learning (specifically for social dilemmas). For example:

Molleman, L., Van den Berg, P., & Weissing, F. J. (2014). Consistent individual differences in human social learning strategies. Nature Communications, 5, 3570.
Molleman, L., & Gächter, S. (2018). Societal background influences social learning in cooperative decision making. Evolution and Human Behavior, 39(5), 547-555.
Lamba, S. (2014). Social learning in cooperative dilemmas. Proceedings of the Royal Society B: Biological Sciences, 281(1787), 20140417.
Replace the term ‘conveyed’ with ‘used’.

- The term was replaced (L 45).
- We now elaborate more on social learning strategies (L 41-47). Specifically, the suggested body of literature is now mentioned in the discussion (L 259-261).

Line 42 – For clarity, perhaps explain how social learning to adjust cooperation could be beneficial first before moving on to relate it to cleaner fish.

- We substantially expanded on this point, and describe potential benefits of social learning in cooperative contexts in a broader sense (L 50-60)

Line 50 – Do cleaner fish then respond to this by increasing their cooperation with that individual? Or do they permanently terminate their interaction?

- Both leaving and chasing mark the termination of the interaction by the client, and lead to more cooperative behaviour by cleaners in future interactions. We modified the sentence to make this clearer (L 66-71).

Line 55 – you can refer to the social learning strategies/transmission bias literature to validate the assumptions of social learning in juvenile fish (ie) copy when ignorant, copy when asocial learning is costly etc. see Kendal RL, Boogert NJ, Rendell L, Laland KN, Webster M & Jones PL. (2018). Social Learning Strategies: Bridge-building between fields. Trends in Cognitive Sciences 22(7): 651-665.

- Thank you for this suggestion. We now elaborate on this more at the previous paragraph, specifying conditions under which social learning can be favoured (L 55-60).

Line 72 – it would be useful here to explain briefly what the ‘evasive response’ actually was (ie) how did the plate move (where to?), did it ‘chase’ the cleaner fish?

- As response type differed between experiments and treatments, we explain this more elaborately in the description of each specific experiment.

It would be useful to give the rationale for the three different experiments before moving onto the results.

- We agree that this was missing and added the rationale of the three experiments to the text (L 96-108).

Results

Line 80 – As described here the control does not sound like a true control? Everything was not held constant except the adult interaction with the plate (ie) the plate/client seems not to move in the control. However, on line 234-6 we discover that the control was very good. Please give this information here.

- We adjusted the sentence so that it would contain the additional information and hope that it is now clearer (L 116-118).

Line 87-91 – The explanation of the results here is not clear. How were their preferences influenced? What is flake concentration? What is ‘feeding adjustment’?

- Yes, we realize that this part was less clear. We substantially modified both the analysis and the text, and hope that it is now easier to follow (L 127-134).

Line 98 – not clear how this makes 4 treatments, rather than 6?

- The social learning experiment was comprised of 4 treatments, and those are the ones to which the text refers. Details regarding the individual learning phase (the additional 2 treatments) appear in the supplementary information (Supplementary Note - we now mention this more explicitly, see L 141-143).

Line 99 – how could observer fish distinguish between tolerant and responsive plates? Were they different colours/sizes/shapes? Was the order in which fish experienced treatments counterbalanced? If not does this invalidate the results?

- The plates differed in colours and patterns, and the order of the treatments was counterbalanced between cleaners. We now added this to the information provided in the results section (L 144, 146).

Line 103 – was there not also a control where the plate did not move (ie) control for ‘tolerant’ plate.

- Yes. Each plate in the control treatment would leave using the same movement pattern that would be used in the observer treatment (i.e. tolerant plates would leave gently, fleeing plates were removed quickly and punishing plate would be moved in the aquarium, in a way similar to the movements used when punishing the cleaners in the observer treatment). We now state that “both observation time and plate movement (gentle leaving/fast fleeing/simulated punishing) were matched to the observer treatments” (L 148-151).

Line 107 – confusingly written “only cleaners of the observer treatments...” Do you mean only following ‘observer trials’ as each fish received each of the 4 treatments (according to line 98)?

- We modified this sentence to “only following trials of the 'observer' treatments the cleaners significantly preferred the tolerant plates” (L 162-165).

Line 115 – state what you actually mean by ‘blindly’? Do you just mean copying of an arbitrary preference?

- We rewrote the sentence. It now states that “In the third experiment, we tested in how far juvenile cleaners would copy whichever behaviours they happen to observe, by exposing them to social information regarding adult cleaners’ arbitrary feeding preferences” (L 166-168).

Line 116 – was this a between subjects design?

Line 119 – was the preferred plate counterbalanced across observers?

- The answer to both questions is yes. We now state this explicitly in the results section (L 180-181, 183-184).

Line 124 – Consider rephrasing for clarity. Perhaps just “We would expect these preferences to be copied”.

- We rephrased the sentence according to the reviewer’s suggestion (L 188-189).

Line 135 – The way this sentence is framed slightly implies that the fish learned about outcomes instead of copying behaviour patterns. This and the proceeding discussion could be rephrased to more clearly emphasise that cleaner fish seemed selective in which behaviours they copied. They only copied behaviour patterns that resulted in tangible outcomes, which is interesting but underplayed.

- Thank you for this suggestion. We now further discuss the fact the cleaners didn't copy (L 218-230). We are a bit careful with the interpretation of their learning as selective, since there are other possible explanations to the negative results of experiment 3 (see supplementary discussion, pages 6-7, L109-131).

Discussion

Line 145 – but were they actually given a choice of plates at test? This needs clarifying in the results section.

- Testing of the juveniles is now described in more detail in the results section (L 151-156)

Line 149 – the authors should consider and compare with other literature that has considered payoff based social learning in animals eg.

Vale, G, Flynn, EG., Kendal, JR., Rawlings, B, Hopper LM., Schapiro SJ., Lambeth SP. & Kendal RL. (2017). Testing differential use of payoff-biased social learning strategies in children and chimpanzees. Proceedings of the Royal Society B Bono AEJ, Whiten A., van Schaik C., Krützen M., Eichenberger F., Schneider A., van de Waal E., 2018. Payoff- and Sex-Biased Social Learning Interact in a Wild Primate Population. Current Biology, 28 (17) pp. 2800-2805.e4. Peer-reviewed.

- We now further discuss payoff based social learning, while specifically referring to both papers (lines 249-259).

Line 155 – It would be helpful to consider specific examples. The major alternative which comes to mind is the Snowdrift game.

Doebeli, M., & Hauert, C. (2005). Models of cooperation based on the Prisoner's Dilemma and the Snowdrift game. Ecology letters, 8(7), 748-766.

- In the introduction, we suggest that such learning may be relevant when there is enough variation – multiple partners, variation in strategies, etc (lines 50-54). In the snowdrift game, the available strategies are discrete, in which case we suspect that individual learning may suffice for reaching effective solutions.

Line 159 – Consider using a simpler word rather than “hitherto”. Perhaps just “currently”.

- We adjusted the phrasing (L 248).

Line 165 – “have an equivalent effect AS CHASERS on juveniles”?

- What we meant here is that our experiment shows a pattern that is similar to theoretical and empirical work in the human literature, whereby the use of negative responses to cheating can affect the behaviour of bystanders. We rephrased the argument (L 268-273).

Line 174-5 - This could be clearer. Do you mean punishment increases cooperation in eavesdroppers but reduces the chance that they choose to interact with the punisher they observed? Please clarify

- Yes, this is precisely what we meant. We modified the sentence following the reviewer's comment: "using partner control mechanisms to respond negatively to cheating can act as a double-edged sword, favourably increasing cooperation in eavesdroppers but also reducing the chance of being chosen as interaction partner" (L 276-279).

In general there is no discussion of ecological validity of the studies. For example, could an explanation for the lack of social learning of an apparent arbitrary preference for one coloured plate over another (experiment 3) be explained by it not representing the differences between clients effectively? What cues do cleaner fish use to distinguish clients, perhaps size or pattern is important and this did not vary between the two plates.

- As noted above, we repeatedly mentioned ecological validity in other sections of the manuscript (L 83-95, 123-124, 134-136, 171-173, 193-197). We now further discuss the ecological validity and potential explanations for the fact that we find no evidence for copying (see Supplementary Discussion pages 6-7). With respect to the reviewer's specific question: clients in the reef vary substantially in a range of visual features, including size, shape, colour, and pattern, and cleaners are likely to rely on these features for distinguishing and choosing between different client types. In lab experiments, previous studies indicate that cleaners can distinguish between plates based on any of these features, including monochromatic plates. Furthermore, the adults in our experiments could be trained to prefer one of the plates over the other (Supplementary Discussion, page 7, L127-128). Thus, it seems unlikely to us that a difficulty to use colours as meaningful cues prevented the cleaners from learning to prioritize one plate over the other. We do suggest that not seeing the food itself, could potentially have an effect.

Materials and Methods

Line 191-194 – unnecessary repetition

Line 213 – Typo, missing space (or missing reference) in brackets.

- Thank you for drawing our attention to these. Both were corrected.

Line 214 – who/what “didn't respond to prawn consumption”?

- In this case, we meant that the plates didn't respond. We modified the sentence to: "In contrast, juvenile cleaners were accustomed to feeding on

plates containing both food types **that** did not respond to prawn consumption” (L 321).

Line 238 – the repeated 15 trials at test (with the plates responding as during observation phase) indicate that individual learning could easily have taken place. What were the results of the very first trial? This is the one for which no individual learning could have yet obscured results? Is there the same problem with experiment 2 (Line 298)?

- Individual learning in itself could not have explained our results, but its potential role should have been addressed and we thank the reviewer for raising this issue. It is now mentioned in the results section (L 120-124, 129-131, 151-156, 169-170), and addressed in detail in the methods (L 352-364, 439-443).

As the reviewer points out, the cleaners could learn through individual learning during our tests, and this is true for all of our social learning experiments. We believe that this design captures the way juvenile cleaners’ social learning might operate in natural settings, and is more realistic than the possible alternative design, in which the plates do not respond at all when the observer cleaners get the chance to interact with them. It is also consistent with the view of social learning as a biasing of individual learning processes by social stimuli (e.g. Galef & Giraldeau 2001, Leadbeater 2015, Truskanov & Lotem 2015, 2017).

While analysis of the first trials in our experiments would not show clear differences between treatment groups. We do not think that this implies that the differences between treatments are caused by individual learning, but simply that one trial comparisons would lack statistical power. In itself, individual learning could not have explained the effects of the social observation in the two experiments. As cleaners of both control and observation groups can apply individual learning during the tests, reliance on such learning would be expected to diminish the effect of our experimental treatments - if the cleaners can easily learn about plate responses by themselves, we wouldn’t find clear differences between cleaners in observer and control treatments, as the control individuals would quickly catch up. Thus, the significant effects of observation in our first two experiments, is exhibited despite the potential for individual learning and not because of it.

Line 241: what qualified as a ‘flake item’ being consumed? Did the fish have to nibble at it or consume it entirely for it to be counted as eaten?

- We considered food consumptions as instances in which the cleaner approached a food item, and targeted it by attaching its mouth to the plate at the item’s location (we added a clarification in the text, see (L 351-352)).

Line 245: experiment 2 is described as having 4 treatments in the Results (this needs clarifying).

- This experiment was divided into two phases: an individual learning phase comprised of 2 treatments (the results of which are reported in the SI) and a social learning phase comprised of 4 treatments. We rewrote this section and

hope that it is now clearer (L 367-375).

Line 322 – Why is it beneficial for the prawns to be on the back of the plates?

- The food was placed on the back of the plate to ensure that the cleaners would not be able to see the food itself before making a choice (i.e. forcing them to base their choices on the colour cues rather than just approaching the visible food). This design further allowed us to pre-train the demonstrators with different reward regimes. We added a more thorough explanation in the text (L 450-455), and also discuss the implications of this in the Supplementary Discussion (page 7, L125-131).

Statistical Analysis

Line 344 – For those unfamiliar, it would be useful to briefly explain AIC and how it is used to evaluate models.

- Following the comments of reviewer #2, we modified the analysis and no longer use model selection.

Line 381 – May be useful to explain the quasibinomial distribution and how it is different to a regular binomial distribution.

- We now explain this in the text (L 551-553)

Reviewers' Comments:

Reviewer #1:

Remarks to the Author:

The MS is greatly improved since the last version. I have no further major issues to raise.

There is a full-stop missing in line 234.

L430: language??

Culum Brown

Reviewer #2:

Remarks to the Author:

I first want to apologise for the delay in returning this review I understand that it is very difficult to wait for reviewers to get their act together but in this case just a number of factors came together and I've not been able to fulfil many of my commitments in a timely fashion.

However I've now had a chance to review the changes the authors have made in response to my comments and I am very happy with the way they have done so - where they have defended their positions I am persuaded to give way, since they have taken my main points onboard in a very positive way, overhauling their analysis significantly - I think the paper is stronger now and ready to be published.

Response to referees NCOMMS-19-19124 manuscript

Dear referees,

We would like to thank you again for the positive and constructive comments in the review process, and were delighted to hear that you found the revised version suitable for publication. Below is an answer to the specific comments made following our revision of the manuscript.

Sincerely yours,

Dr. Noa Truskanov

REVIEWERS' COMMENTS:

- Our response

Reviewer #1 (Remarks to the Author):

The MS is greatly improved since the last version. I have no further major issues to raise.

- Thank you, this was nice to hear!

There is a full-stop missing in line 234.

- Thank you for drawing our attention – we now added it.

L430: language??

- We modified the phrasing.

Culum Brown

Reviewer #2 (Remarks to the Author):

I first want to apologise for the delay in returning this review I understand that it is very difficult to wait for reviewers to get their act together but in this case just a number of factors came together and I've not been able to fulfil many of my commitments in a timely fashion.

However I've now had a chance to review the changes the authors have made in response to my comments and I am very happy with the way they have done so - where they have defended their positions I am persuaded to give way, since they have taken my main points onboard in a very positive way, overhauling their analysis significantly - I think the paper is stronger now and ready to be published.

- We are glad to hear that you agree with the changes we made, and would like to thank you again for the constructive feedback.